# Incorporating Horizontal Connections in Convolution by Spatial Shuffling

Convolutional Neural Networks (CNNs) are composed of multiple convolution layers and show elegant performance in vision tasks. The design of the regular convolution is based on the Receptive Field (RF) where the information within a specific region is processed. In the view of the regular convolution's RF, the outputs of neurons in lower layers with smaller RF are bundled to create neurons in higher layers with larger RF. As a result, the neurons in high layers are able to capture the global context even though the neurons in low layers only see the local information. However, in lower layers of the biological brain, the information outside of the RF changes the properties of neurons. In this work, we extend the regular convolution and propose spatially shuffled convolution (ss convolution). In ss convolution, the regular convolution is able to use the information outside of its RF by spatial shuffling which is a simple and lightweight operation. We perform experiments on CIFAR-10 and ImageNet-1k dataset, and show that ss convolution improves the classification performance across various CNNs.

## 1 Introduction

Convolutional Neural Networks (CNNs) and their convolution layers (Fukushima, 1980; Lecun et al., 1998) are inspired by the finding in cat visual cortex (Hubel & Wiesel, 1959) and they show the strong performance in various domains such as image recognition (Krizhevsky et al., 2012; Simonyan & Zisserman, 2015; He et al., 2016), natural language processing (Gehring et al., 2017), and speech recognition (Abdel-Hamid et al., 2014; Zhang et al., 2016). A notable characteristic of the convolution layer is the Receptive Field (RF), which is the particular input region where a convolutional output is affected by. The units (or neurons) in higher layers have larger RF by bundling the outputs of the units in lower layers with smaller RF. Thanks to the hierarchical architectures of CNNs, the units in high layers are able to capture the global context even though the units in low layers only see the local information.

It is known that neurons in the primary visual cortex (i.e., V1 which is low layers) change the self-properties (e.g., the RF size (Pettet & Gilbert, 1992) and the facilitation effect (Nelson & Frost, 1985)) based on the information outside of the RF (D.Gilbert, 1992). The mechanism is believed to originate from (1) feedbacks from the higher-order area (Iacaruso et al., 2017) and (2) intra-cortical horizontal connections (D.Gilbert, 1992). The feedbacks from the higher-order area convey broader-contextual information than the neurons in V1, which allows the neurons in V1 to use the global context. For instance, Gilbert & Li (2013) argued that the feedback connections work as attention. Horizontal connections allow the distanced neurons in the layer to communicate with each other and are believed to play an important role in visual contour integration (Li & Gilbert, 2002) and object grouping (Schmidt et al., 2006).

Though both horizontal and feedback connections are believed to be important for visual processing in the visual cortex, the regular convolution ignores the properties of these connections. In this work, we particularly focus on algorithms to introduce the function of horizontal connections for the regular convolution in CNNs. We propose spatially shuffled convolution (ss convolution), where the information outside of the regular convolution's RF is incorporated by spatial shuffling, which is a simple and lightweight operation. Our ss convolution is the same operation as the regular convolution except for spatial shuffling and requires no extra learnable parameters. The design of ss convolution is highly inspired by the function of horizontal connections. To test the effectiveness of the information outside of the regular convolution's RF in CNNs, we perform experiments on CIFAR-10 (Krizhevsky, 2009) and ImageNet 2012 dataset (Russakovsky et al., 2015) and show that ss convolution improves the classification performance across various CNNs. These results indicate that the information outside of the RF is useful when processing local information. In addition, we

conduct several analyses to examine why ss convolution improves the classification performance in CNNs and show that spatial shuffling allows the regular convolution to use the information outside of its RF.

## 2 RELATED WORK

### 2.1 VARIANTS OF CONVOLUTION LAYERS AND NEURAL MODULES

There are two types of approaches to improve the Receptive Field (RF) of CNNs with the regular convolution: broadening kernel of convolution layer and modulating activation values by self-attention.

**Broadening Kernel:**

The atrous convolution (Holschneider et al., 1989; Yu & Koltun, 2016) is the convolution with the strided kernel. The stride is not learnable and given in advance. The atrous convolution can have larger RF compared to the regular convolution with the same computational complexity and the number of learnable parameters.

The deformable convolution (Dai et al., 2017) is the atrous convolution with learnable kernel stride that depends on inputs and spatial locations. The stride of the deformable convolution is changed flexibly unlike the atrous convolution, however, the deformable convolution requires extra computations to calculate strides.

Both atrous and deformable convolution contribute to broadening RF, however, it is not plausible to use the pixel information at a distant location when processing local information. Let us consider the case that the information of $p$ pixels away is useful for processing local information at layer $l$. In the simple case, it is known that the size of the RF grows with $k\sqrt{n}$ where $k$ is the size of the convolution kernel and $n$ is the number of layers (Luo et al., 2016). In this case, the size of kernel needs to be $\frac{p}{\sqrt{n}}$ and $k$ is around 45 when $p = 100$ and $l = 5$. If the kernel size is $3 \times 3$, then the stride needs to be 21 across layers. Such large stride causes both the atrous and the deformable convolution to have a sparse kernel and it is not suitable for processing local information.

**Self-Attention:**

Squeeze and Excitation module (SE module) (Hu et al., 2018) is proposed to modulate the activation values by using the global context which is obtained by Global Average Pooling (GAP) (Lin et al., 2014). SE module allows CNNs with the regular convolution to use the information outside of its RF as our ss convolution does. In our experiments, ss convolution gives the marginal improvements on SEResNet50 (Hu et al., 2018) that is ResNet50 (He et al., 2016) with SE module. This result makes us wonder why ss convolution improves the performance of SEResNet50, thus we conduct the analyses and find that the RF of SEResNet50 is location independent and the RF of ResNet with ss convolution is the location-dependent. This result is reasonable since the spatial information of activation values is not conserved by GAP in SE module. We conclude that such a difference may be the reason why ss convolution improves the classification performance on SEResNet50.

Attention Branch Networks (ABN) (Fukui et al., 2019) is proposed for a top-down visual explanation by using an attention mechanism. ABN uses the output of the side branch to modulate activation values of the main branch. The outputs of the side branch have larger RF than the one of the main branch, thus the main branch is able to modulate the activation values based on the information outside of main branch's RF. In our experiments, ss convolution improves the performance on ABN and we assume that this is because ABN works as like feedbacks from higher-order areas, unlike ss convolution that is inspired by the function of horizontal connections.

### 2.2 UTILIZATION OF SHUFFLING IN CNNS

ShuffleNet (Zhang et al., 2017) is designed for computation-efficient CNN architecture and the group convolution (Krizhevsky et al., 2012) is heavily used. They shuffle the channel to make cross-group information flow for multiple group convolution layers.

The motivation of using shuffling between ShuffleNet and our ss convolution is different. On the one hand, our ss convolution uses spatial shuffling to use the information from outside of the regular

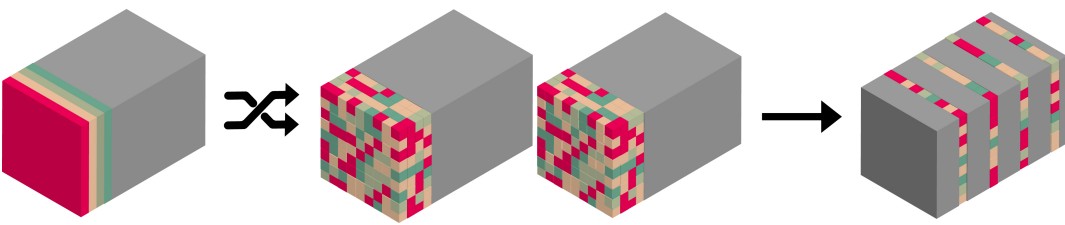

(a) Spatial Shuffling for Convolution (Eqn. 2)      (b) Spatial Shuffling for Group Convolution (Eqn. 3)

Figure 1: The overview of spatial shuffling operarion.

convolution's RF. On the other hand, the channel shuffling in ShuffleNet does not broaden RF and not contribute to use the information outside of the RF.

## 3 METHOD

In this section, we introduce spatially shuffled convolution (ss convolution).

### 3.1 SPATIALLY SHUFFLED CONVOLUTION

Horizontal connections are the mechanism to use information outside of the RF. We propose ss convolution to incorporate this mechanism into the regular convolution, which consists of two components: spatial shuffling and regular convolution. The shuffling is based on a permutation matrix that is generated at the initialization. The permutation matrix is fixed while training and testing.

Our ss convolution is defined as follows:

$$y_{i,j} = \sum_{c}^{C} \sum_{\Delta i, \Delta j \in R} w_{c, \Delta i, \Delta j} \cdot P(x_{c, i+\Delta i, j+\Delta j}), \tag{1}$$

$$P(x_{c,i,j}) = \begin{cases} \pi(x_{c,i,j}), & c \leq \lfloor \alpha C \rfloor, \\ x_{c,i,j}, & \text{otherwise.} \end{cases} \tag{2}$$

$R$ represents the offset coordination of the kernel. For examples, the case of the $3 \times 3$ kernel is $R = \{(-1,-1), (-1,0), (-1,1), (0,-1), (0,0), (0,1), (1,-1), (1,0), (1,1)\}$. $x \in \mathbb{R}^{C \times I \times J}$ is the input and $w \in \mathbb{R}^{C_w \times I_w \times J_w}$ is the kernel weights of the regular convolution. In Eqn. (2), the input $x$ is shuffled by $P$ and then the regular convolution is applied. Fig. 1-(a) is the visualization of Eqn. (2). $\alpha \in [0, 1]$ is the hyper-parameter to control how many channels are shuffled. If $\lfloor \alpha C \rfloor = 0$, then ss convolution is same as the regular convolution. At the initialization, we randomly generate the permutation matrix $\pi \in \{0, 1\}^{m \times m}$ where $\sum_{i=1}^{m} \pi_{i,j} = 1$, $\sum_{j=1}^{m} \pi_{i,j} = 1$ and $m = I \cdot J \cdot \lfloor \alpha C \rfloor$[1]. The generated $\pi$ at the initialization is fixed for training and testing.

The result of CIFAR-10 across various $\alpha$ is shown in Fig. 2. The biggest improvement of the classification performance is obtained when $\alpha$ is around 0.06.

### 3.2 SPATIALLY SHUFFLED GROUP CONVOLUTION

The group convolution (Krizhevsky et al., 2012) is the variants of the regular convolution. We find that the shuffling operation of Eqn. 2 is not suitable for the group convolution. ResNeXt (Xie et al., 2017) is CNN to use heavily group convolutions and Table 1 shows the test error of ResNeXt in CIFAR-10 (Krizhevsky, 2009). As can be seen in Table 1, the improvement of the classification performance is marginal with Eqn. 2. Thus, we propose the spatial shuffling for the

---

[1]We implement Eqn. 2 by indexing, thus we hold $m$ long int instead of $m \times m$ binary matrix. The implementation of ss convolution is shown in Appendix A.2.

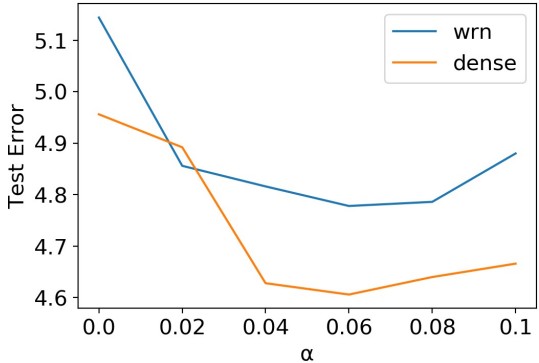

Figure 2: The result of CIFAR-10 across various $\alpha$. WRN 16-4 (Zagoruyko & Komodakis, 2016) and DenseNet-BC 12-100 (Huang et al., 2017) are used. The test error at the last epoch is used and the reported results are averaged out of 5 runs.

| model | method | test error (%) |
|---|---|---|
| ResNeXt-29 (2x64d) | Conv | 4.3 |
| | SS Conv w/ Eqn. (2) | 4.2 |
| | SS Conv w/ Eqn. (3) | **3.9** |

Table 1: The result of CIFAR-10. The test error at the last epoch is used and the reported results are averaged out of 5 runs.

group convolution as follows:

$$P(x_{c,i,j}) = \begin{cases} \pi(x_{c,i,j}), & 0 \equiv C \mod \lfloor \frac{1}{\alpha} \rfloor, \\ x_{c,i,j}, & \text{otherwise.} \end{cases} \quad (3)$$

Eqn. 3 represents that the shuffled parts are interleaved like the illustration in Fig. 1-(b). As can be seen in Table 1, ss convolution with Eqn. 3 improves the classification performance of ResNeXt.

## 4 EXPERIMENTS

### 4.1 PREPARATIONS

We use CIFAR-10 (Krizhevsky, 2009) and ImageNet-1k (Russakovsky et al., 2015) for our experiments.

**CIFAR-10.** CIFAR-10 is the image classification dataset. There are 50000 training images and 10000 validation images with 10 classes. As data augmentation and preprocessing, translation by 4 pixels, stochastic horizontal flipping, and global contrast normalization are applied onto images with $32 \times 32$ pixels. We use three types of models of WRN 16-4 (Zagoruyko & Komodakis, 2016), DenseNet-BC 12-100 (Huang et al., 2017) and ResNeXt 2-64d (Xie et al., 2017).

**ImageNet-1k.** ImageNet-1k is the large scale dataset for the image classification. There are 1.28M training images and 50k validation images with 1000 classes. As data augmentation and preprocessing, resizing images with the scale and aspect ratio augmentation and stochastic horizontal flipping are applied onto images. Then, global contrast normalization is applied to randomly cropped images with $224 \times 224$ pixels. In this work, we use ResNet50 (He et al., 2016), DenseNet121 (Huang et al., 2017), SEResNet50 (Hu et al., 2018) and ResNet50 with ABN (Fukui et al., 2019) for ImageNet-1k experiments.

**Implementation Details.** As the optimizer, we use Momentum SGD with momentum of 0.9 and weight decay of $1.0 \times 10^{-4}$. In CIFAR-10, we train models for 300 epochs with 64 batch size. In ImageNet, we train models for 100 epochs with 256 batch size. In CIFAR-10 and ImageNet, the learning rate starts from 0.1 and is divided by 10 at 150, 250 epochs and 30, 60, 90 epochs, respectively.

| model | method | test error (%) |
|---|---|---|
| WRN 16-4 | Conv | 5.1 |
| | SS Conv | **4.8** |
| DenseNet-BC | Conv | 5.0 |
| | SS Conv | **4.6** |
| ResNeXt-29 (2x64d) | Conv | 4.3 |
| | SS Conv | **3.9** |

Table 2: The result of CIFAR-10. The test error at the last epoch is used and the reported results are averaged out of 5 runs.

| model | method | top-1 err (%) |
|---|---|---|
| ResNet50 | Conv | 23.5 |
| | SS Conv | **23.0** |
| ResNet50 w/ ABN | Conv | 22.9 |
| | SS Conv | **22.6** |
| SEResNet50 | Conv | 22.7 |
| | SS Conv | **22.6** |
| DenseNet121 | Conv | 24.5 |
| | SS Conv | **24.1** |
| ResNeXt-50 (32x4d) | Conv | 22.4 |
| | SS Conv | **22.0** |

Table 3: Top-1 error on ImageNet-1k validation dataset. The top-1 error at the last epoch is used and the reported results are averaged out of 3 runs.

| model | method | inference speed (ms) |
|---|---|---|
| ResNet50 | Conv | 1.44 |
| | SS Conv | 1.66 |
| SEResNet50 | Conv | 2.11 |

Table 4: The inference speed per image is shown. We evaluate models on single GeForce GTX 1080 Ti cards with CUDA 10.0 environment. The implementation is based on PyTorch (Paszke et al., 2017) ver 1.0.1.post2. The results are averaged out of 1000 runs. The batch size is 256 and the size of input is $224 \times 224 \times 3$. $\alpha$ of ss convolution is $0.04$ that is same $\alpha$ reported in Table 3.

In our experiments, we replace all regular convolutions with ss convolutions except for downsampling layers, and use single $\alpha$ across all layers. We conduct grid search of $\alpha \in \{0.02, 0.04, 0.06\}$ and $\alpha$ is decided according to the classification performance on validation dataset.

## 4.2 RESULTS

We replace all regular convolutions with ss convolutions to investigate whether the information outside of the regular convolution's RF contributes to improving the generalization ability. The results are shown in Table 2 and 3. As can be seen in Table 2 and 3, ss convolution contributes to improve the classification performance across various CNNs except for SEResNet50 that shows marginal improvements. The detailed analysis of the reason why ss convolution gives the marginal improvements in SEResNet50 is shonw in Sec. 5

Since $\alpha \in \{0.02, 0.04, 0.06\}$ is small, the small portion of the input are shuffled, thus ss convolution improves the classification performance with small amount of extra shuffling operations and without extra learnable parameters. The inference speed is shown in Table 4 and ss convolution make the inference speed $1.15$ times slower in exchange for $0.5\%$ improvements in ImageNet-1k dataset. The more efficient implementation[2] may decrease the gap of the inference speed between the regular convolution and ss convolution.

## 5 ANALYSIS

In this section, we demonstrate two analysis to understand why ss convolution improves the classification performance across various CNNs: the receptive field (RF) analysis and the layer ablation experiment.

---

[2]Our implementation is shown in Appendix A.2.

| layer name | output size | blue box size | structure |
|:---:|:---:|:---:|:---:|
| input | $224 \times 224$ | $32 \times 32$ | |
| conv1 | $112 \times 112$ | $16 \times 16$ | $7 \times 7$, 64, stride 2 |
| | $56 \times 56$ | $8 \times 8$ | $3 \times 3$ max pool, 64, stride 2 |
| conv2_x | $56 \times 56$ | $8 \times 8$ | $\begin{bmatrix} 1 \times 1, 64 \\ 3 \times 3, 64 \\ 1 \times 1, 256 \end{bmatrix} \times 3$ |
| conv3_x | $28 \times 28$ | $4 \times 4$ | $\begin{bmatrix} 1 \times 1, 128 \\ 3 \times 3, 128 \\ 1 \times 1, 512 \end{bmatrix} \times 4$ |
| conv4_x | $14 \times 14$ | $2 \times 2$ | $\begin{bmatrix} 1 \times 1, 256 \\ 3 \times 3, 256 \\ 1 \times 1, 1024 \end{bmatrix} \times 6$ |
| conv5_x | $7 \times 7$ | $1 \times 1$ | $\begin{bmatrix} 1 \times 1, 512 \\ 3 \times 3, 512 \\ 1 \times 1, 2048 \end{bmatrix} \times 3$ |
| | $1 \times 1$ | | average pool, 1000-d fc, softmax |

Table 5: The structure of ResNet50 (He et al., 2016).

**Receptive Field Analysis.** We calculate the RF of SEResNet50, ResNet50 with ss convolution and the regular convolution. The purpose of this analysis is to examine whether ss convolution contributes to use the information outside of the regular convolution's RF.

**Layer Ablation Experiment.** The layer ablation experiment is conducted to know which ss convolution influences the model prediction. In the primary visual cortex, the neurons change self-properties based on the information outside of RF, thus we would like to investigate whether spatial shuffling in low layers contribute to predictions or not.

Our analyses are based on ImageNet-1k pre-trained model and the structure of ResNet50 (i.e., the base model for analysis) is shown in Table 5.

## 5.1 DOES SPATIAL SHUFFLING CONTRIBUTE TO USE THE INFORMATION FROM OUTSIDE OF RECEPTIVE FIELD?

In our analysis, we calculate the RF to investigate whether ss convolution uses the information outside of the regular convolution's RF. The receptive field is obtained by optimization as follows:

$$\mathbf{R}^* = \arg \min_{\mathbf{R}} \| (M \cdot \phi_l(\sigma(\mathbf{R}) \cdot x) - \phi_l(x) \|_2^2 + \beta \|\sigma(\mathbf{R})\|_1 . \tag{4}$$

$x \in \mathbb{R}^{C \times I \times J}$ is input, and $\mathbf{R} \in \mathbb{R}^{C \times I \times J}$ is the RF to calculate and learnable. $\sigma$ is sigmoid function, thus $0 \leq \sigma(\mathbf{R}) \leq 1$. $\phi_l$ is the outpus of the trained model at the layer $l$.

We call the first term in Eqn. 4 as the local perceptual loss. It is similar to the perceptual loss (Johnson et al., 2016), and the difference is the dot product of $M \in \{0, 1\}^{C \times I \times J}$ that works as masking. $M$ is the binary mask and $M_{cij} = 1$ if $96 \leq i, j \leq 128$, otherwise $M_{cij} = 0$ in our analysis. In other words, the values inside the blue box in Fig 3 are the part of $M_{cij} = 1$. The local perceptual loss minimizes the distance of feature on the specific region between $\sigma(\mathbf{R}) \cdot x$ and $x$. The 2nd term is the penalty to evade the trivial case such as $\sigma(\mathbf{R}) = \mathbb{1}$. In layerwise and channelwise RF anlysis, we use $\beta$ of $1.0 \times 10^{-6}$ and $1.0 \times 10^{-12}$, respectively.

We use Adam optimizer (Kingma & Ba, 2015) to calculate $\mathbf{R}^*$. As the hyper-parameter, lr, $\beta_1$, and $\beta_2$ are $0.1, 0.9, 0.99$, respectively. The high lr is used since its slow convergence. The batch size is 32 and we stop the optimization after 10000 iterations. $x$ is randomly selected from ImageNet-1k training images. The data augmentation and preprocessing are applied as the same procedure in Sec. 4.1.

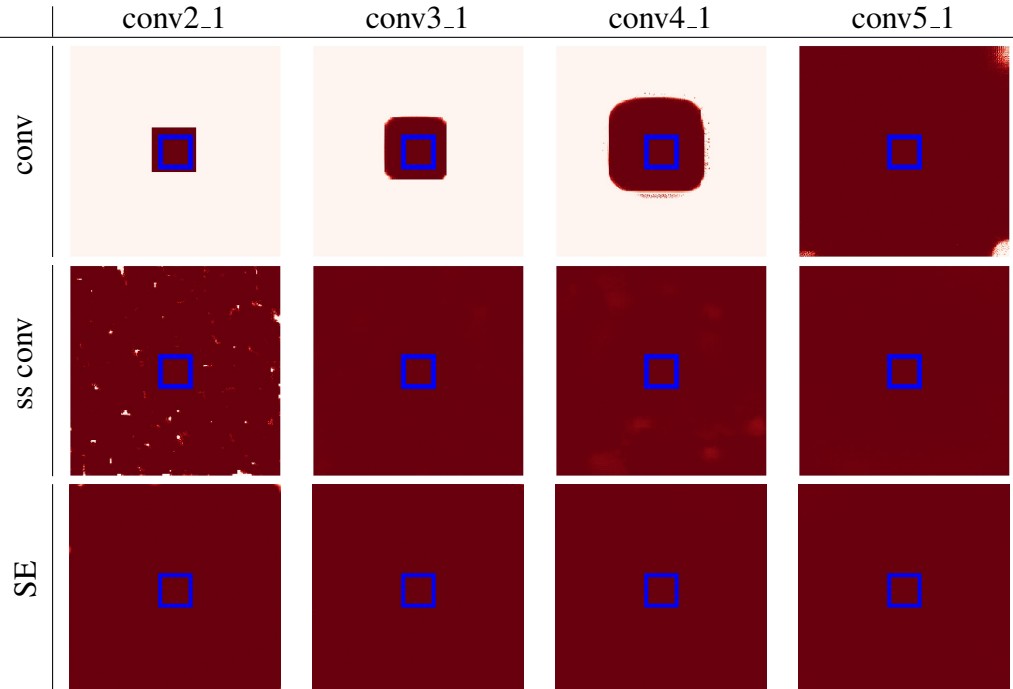

Figure 3: The layerwise RF of ImageNet-1k pre-trained models. The layerwise RF images in the top row are based on ResNet50 with the regular convolution, the ones in the middle row are calculated from ResNet50 with ss convolutions and the ones in the bottom row are generated from SEResNet50. The red color indicates that the pixel there changes features inside blue box, and the white color represents that features are invariant. The name of the layer is described in Table 5. The rest of RFs are shown in Appendix A.1.

**Layerwise Receptive Field.** We calculate the RF for each model and the results are shown in Fig. 3. The top row is the RF of ResNet with the regular convolution, the middle row is the one with ss convolution and the bottom row is the one of SEResNet50. The red color indicates that the value of the pixel there changes features inside the blue box, and the white color represents that features inside the blue box are invariant even if the value of the pixel there is changed.

In the top row of Fig. 3, the RFs of ResNet50 with the regular convolution are shown. The size of RF becomes larger as the layer becomes deeper. This result is reasonable and obtained RFs are in the classical view. If the RF of ResNet50 with ss convolution is beyond the one with the regular convolution, it indicates that ss convolution successfully uses the information outside of the regular convolution's RF.

In the middle and bottom row of Fig. 3 are the RF of ResNet50 with ss convolution and SEResNet50, respevtively. The RFs covers the entire image unlike the RF with the regular convolution. These results indicate that both SE module and ss convolution contributes to use the information outside of the regular convolution's RF.

Fig. 5-(a) shows the size of the RF across layers. The horizontal axis is the name of the layer and the vertical axis represents the size of RF that is calculated as $\frac{\|\sigma(R) \geq 0.5\|_1}{|R|}$ where $\| \ \|_1$ is the L1 norm and $| \ |$ is the total size of matrix. $\frac{\|\sigma(RF) \geq 0.5\|_1}{|RF|}$ is in the range between 0 and 1 and represents the ratio of $\sigma(RF)$ that is bigger than $0.5$. As can be seen in Fig. 5-(a), the size of RFs are consistently almost 1 across layers in ResNet50 with ss convolution and SEResNet50. This result also shows that SE module and the ss convolution contributes to use the information outside of the regular convolution's RF. This may be the reason why ss convolution improves the classification performance on various CNNs. However, these results make us wonder why ss convolution improves marginally the performance of SEResNet50. Further analysis is conducted in channelwise RF analysis.

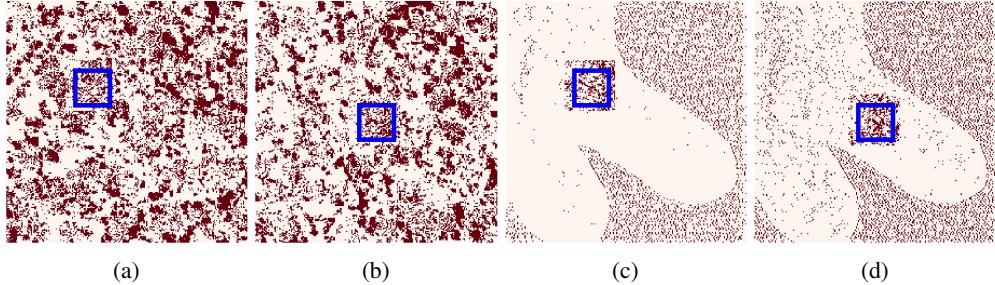

Figure 4: The channelwise RF of ImageNet-1k pre-trained model. The RF is calculated based on the output of the channel 64 in conv2_1. (a)-(b) RF of ResNet50 with ss convolution. (c)-(d) RF of SEResNet50. The red color indicates that the pixel there changes features inside the blue box, and the white color represents that features are invariant. For clarity, we convert the values that satisfy $\sigma(R) \geq 0.1$ into 1.0.

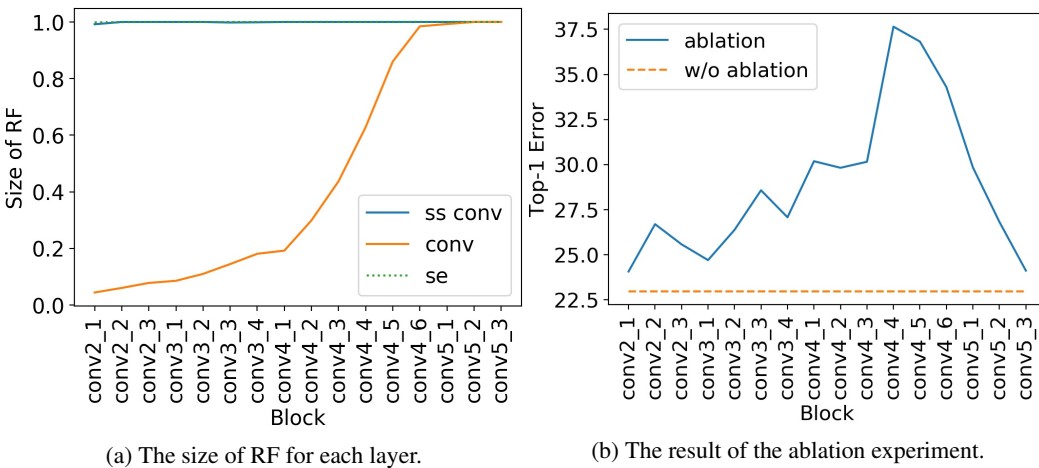

(a) The size of RF for each layer.

(b) The result of the ablation experiment.

Figure 5: The analysis for each layer in ImageNet-1k pretrained models. (a) The horizontal axis is which block is used to calculate the layerwise RF and the vertical axis is the size of RF. (b) The horizontal axis is which block is ablated in ResNet50 with ss convolutions and the vertical axis is the top-1 error in ImageNet-1k validation dataset.

**Channelwise Receptive Field.** Since layerwise RF analysis is based on the RF of the layer, the obtained results have rough directions. We calculate the channelwise RF for more fine-grained analysis. Unlike layerwise RF analysis, $M$ becomes different and we minimize the local perceptual loss on the specific channel. The results are shown in Fig. 3. Fig. 4 (a) and (c) use $M_{cij} = 1$ if $64 \leq i, j \leq 96$ and $c = 64$, otherwise $M_{cij} = 0$. Fig. 4 (b) and (d) use $M_{cij} = 1$ if $96 \leq i, j \leq 128$ and $c = 64$, otherwise $M_{cij} = 0$. Fig. 4 (a)-(b) are the RF of ResNet50 with ss convolution and (c)-(d) are tbe RF of SEResNet50. As can be seen in Fig. 3, the RFs of ResNet50 with ss convolution (i.e., Fig. 4 (a)-(b)) are different when the blue box is shifted, however, the RFs of SEResNet50 (i.e., Fig. 4 (c)-(d)) are similar even if the blue box is shifted. These results indicate that the information outside of the regular convolution's RF is location-independent in SEResNet 50 and location-dependent in ResNet50 with ss convolution. This is reasonable since SE module uses the global average pooling and the spatial information is not conserved. This difference may be the reason why ss convolution marginally improves the classification performance on SEResNet50.

## 5.2 ABLATION STUDY

We conduct layer ablation study to investigate which ss convolutions contribute to the generalization ability. The ablation is done as follows:

$$P(x_{c,i,j}) = \begin{cases} 0, & c \leq \lfloor \alpha C \rfloor, \\ x_{c,i,j}, & \text{otherwise.} \end{cases} \tag{5}$$

Eqn. 5 represents that the activation values of the shuffled parts become 0. The result of the ablation experiment is shown in Fig. 5-(b). Eqn. 5 is applied to all ss convolutions in each block and the biggest drop of the classification performance happens at the ablation of conv4_4. It indicates that it is useful to use the information outside of the regular convolution's RF between the middle and high layers. The classification performance is degraded even if the ablation is applied to the first bottle-neck (i.e., conv2_1). This result implies that the information outside of the regular convolution's RF is useful even at low layers.

## 6 CONCLUSION

In this work, we propose spatially shuffled convolution (ss convolution) to incorporate the function of horizontal connections in the regular convolution. The spatial shuffling is simple, lightweight, and requires no extra learnable parameters. The experimental results demonstrate that ss convolution captures the information outside of the regular convolution's RF even in lower layers. The results and our analyses also suggest that using distant information (i.e., non-local) is effective for the regular convolution and improves classification performance across various CNNs.

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

## A  APPENDIX

### A.1  RESULTS OF LAYERWISE RECEPTIVE FIELD

The receptive fields of all layers are shown in Fig. 6, 7 and 8.

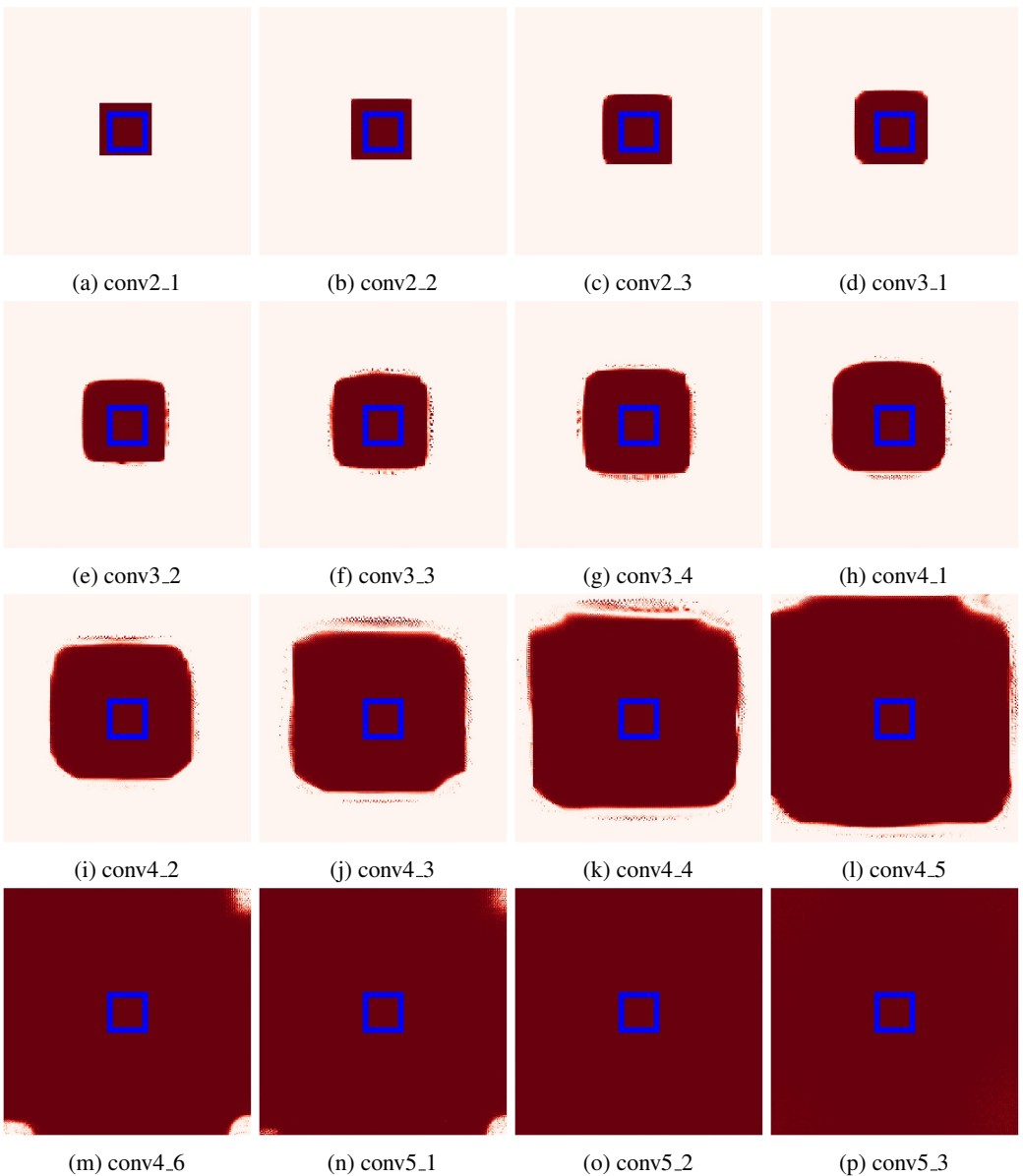

Figure 6: The receptive field of ImageNet-1k pre-trained ResNet50. The red color indicates that the pixel there changes features inside the blue box, and the white color represents that features are invariant even if the pixel there changes the value itself. Those images are the receptive field of all layers and the name of the layer is described in Table 5

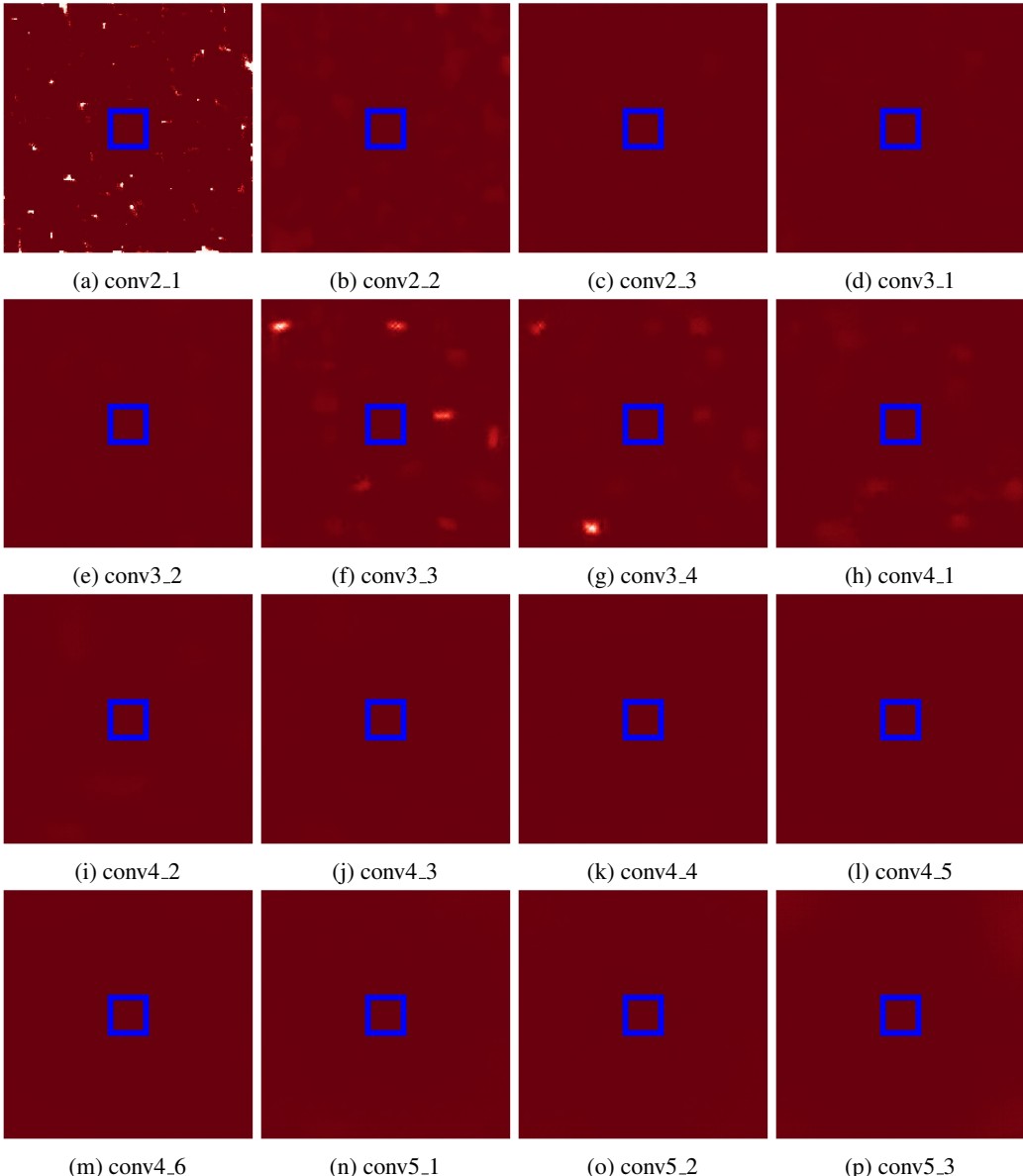

Figure 7: The receptive field of ImageNet-1k pre-trained ResNet50 with ss convolutions. The red color indicates that the pixel there changes features inside the blue box, and the white color represents that features are invariant even if the pixel there changes the value itself. Those images are the receptive field of all layers and the name of the layer is described in Table 5

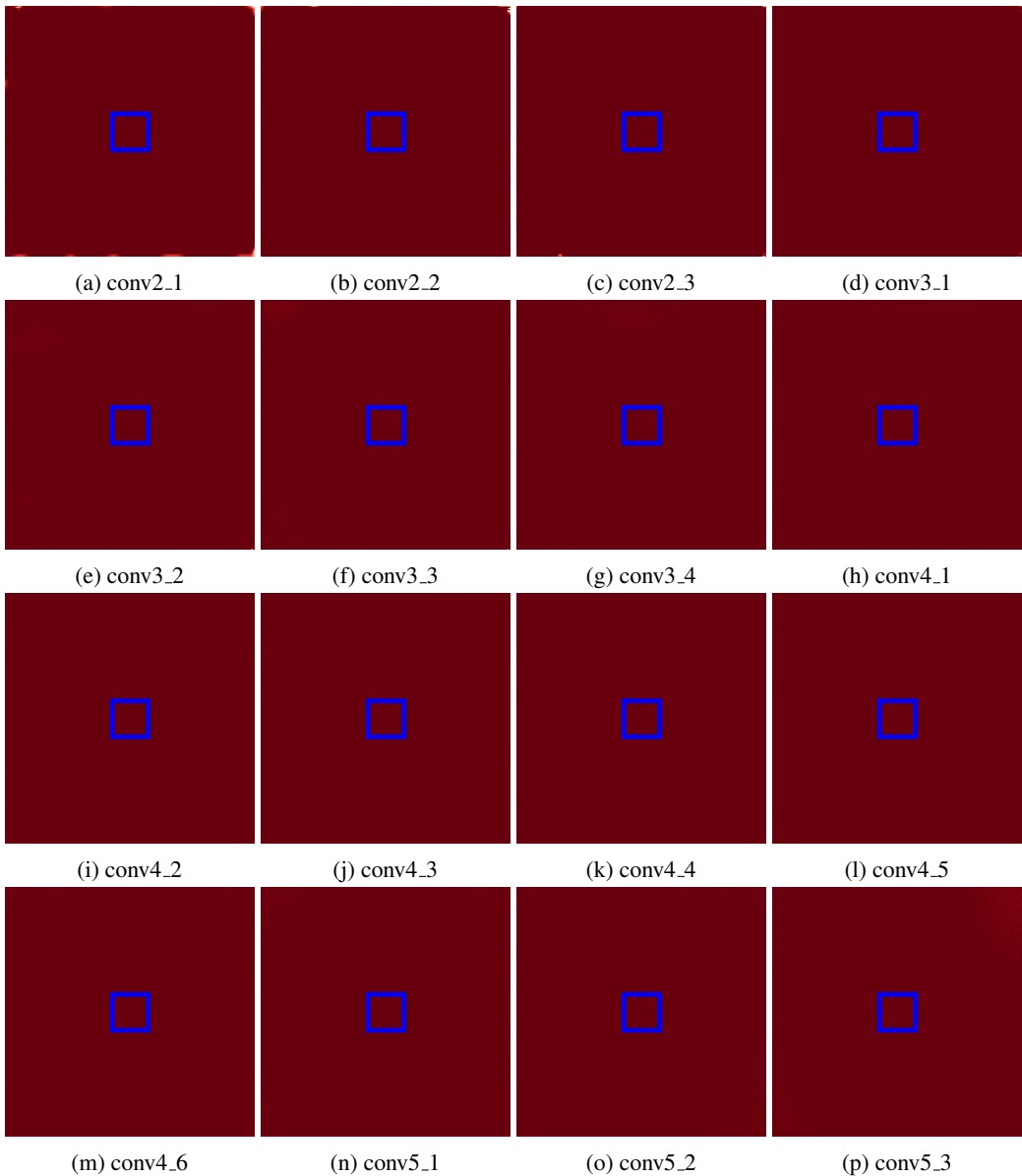

Figure 8: The receptive field of ImageNet-1k pre-trained SEResNet50. The red color indicates that the pixel there changes features inside the blue box, and the white color represents that features are invariant even if the pixel there changes the value itself. Those images are the receptive field of all layers and the name of the layer is described in Table 5

## A.2 EXAMPLE OF CODE

The code of spatially shuffled convolution is shown in Listing 1. It is written in python with Pytorch (Paszke et al., 2017) of 1.0.1.post2 version. The training and model codes will be available online after the review.

```python
import torch

class SSConv2d(torch.nn.Module):

    def __init__(self, in_planes, out_planes, kernel_size=3, stride=1, padding=1, bias=None,
        groups=1, dilation=1, alpha=0.04):
        super(SSConv2d, self).__init__()
        self.conv = torch.nn.Conv2d(in_planes, out_planes, kernel_size=kernel_size, stride=
        stride, padding=padding, groups=groups, dilation=dilation, bias=bias)
        self.alpha, self.groups = alpha, groups

    def create_shuffle_indices(self, x):
        _, in_planes, height, width = x.size()
        self.shuffle_until_here = int(in_planes * self.alpha)
        # if self.shuffle_until_here = 0, then it's exactly same as regular convolution
        if self.shuffle_until_here >= 1:
            self.register_buffer('random_indices', torch.randperm(self.shuffle_until_here *
        height * width))

    @staticmethod
    def _group(shuffled_x, non_shuffled_x):
        batch, ch_ns, height, width = non_shuffled_x.shape
        _, ch_s, _, _ = shuffled_x.shape
        length = int(ch_ns / ch_s)
        residue = ch_ns - length * ch_s
        # shuffled_x is interleaved
        if residue == 0:
            return torch.cat((shuffled_x.unsqueeze(1), non_shuffled_x.view(batch, length, ch_s
        , height, width)), 1).view(batch, ch_ns + ch_s, height, width)
        else:
            return torch.cat((torch.cat((shuffled_x.unsqueeze(1), non_shuffled_x[:, residue:].
        view(batch, length, ch_s, height, width)), 1).view(batch, ch_ns + ch_s - residue, height,
         width), non_shuffled_x[:, :residue]), 1)

    def shuffle(self, x):
        if self.shuffle_until_here >= 1:
            # ss convolution
            shuffled_x, non_shuffled_x = x[:, :self.shuffle_until_here], x[:, self.
        shuffle_until_here:]
            batch, ch, height, width = shuffled_x.size()
            shuffled_x = torch.index_select(shuffled_x.view(batch, -1), 1, self.random_indices
        ).view(batch, ch, height, width)
            if self.groups >= 2:
                return self._group(shuffled_x, non_shuffled_x)
            else:
                return torch.cat((shuffled_x, non_shuffled_x), 1)
        else:
            # regular convolution
            return x

    def forward(self, x):
        if hasattr(self, 'random_indices') is False:
            # create random permutation matrix at initialization
            self.create_shuffle_indices(x)
        # spatial shuffling
        x = self.shuffle(x)
        # regular convolution
        x = self.conv(x)
        return x
```

Listing 1: Implementation of Spatially Shuffled Convolution

