# OpenReview forum: "Incorporating Horizontal Connections in Convolution by Spatial Shuffling"
_ICLR.cc/2020/Conference — Reject_

### Official Review · AnonReviewer3 · 2019-10-25
**Official Blind Review #3**

**Rating:** 3

**Review:**

Summary: The authors extended the regular convolution and proposed spatially shuffled convolution to use the information outside of its RF, which is inspired by the idea that horizontal connections are believed to be important for visual processing in the visual cortex in biological brain. The authors proposed ss convolution for regular convolution and group convolution. The authors tested the proposed ss convolution on multiple CNN models and show improvement of results. Finally, detailed analysis of spatial shuffling and ablation study was conducted.
Strengths: The authors proposed spatially shuffled convolution to use the information outside of its RF. The operation only requires small amount of extra shuffling operations and without extra learnable parameters. The idea is straightforward and easy to understand. I especially like the visualization analysis of the receptive field and the layer ablation study suggested that all layers can benefit from the proposed operation, though more for middle and higher layers.
Weakness: The proposed method seems to be a reasonable alternative for regular convolution, but  By fixing the permutation, not much insight is gained from this technique.
Suggestions: It would be good to include the standard deviation for the results as the final results were average of 5 runs and can help to see if it’s consistently useful. In the current setting, the permutation matrix is generated randomly, considering the proposed method is based on shuffling, how would different permutation affect the performance? In addition, I think it could be helpful if the authors can show more utilities of the proposed method, for example, are the proposed method capable of doing tasks similar to this paper **Learning long-range spatial dependencies with horizontal gated recurrent units**, which argues that the ability of CNNs to learn long-range spatial dependencies is limited by their localized receptive fields. One minor thing, in the main paper, the abbreviation for spatial shuffled convolution (ss convolution) is mentioned multiple times.

**Experience Assessment:**

I have read many papers in this area.

**Review Assessment: Checking Correctness Of Derivations And Theory:**

I assessed the sensibility of the derivations and theory.

**Review Assessment: Checking Correctness Of Experiments:**

I assessed the sensibility of the experiments.

**Review Assessment: Thoroughness In Paper Reading:**

I read the paper thoroughly.

---

### Official Review · AnonReviewer1 · 2019-10-26
**Official Blind Review #1**

**Rating:** 3

**Review:**

[Overview]

In this paper, the authors proposed a shuffle strategy for convolution layers in convolutional neural networks (CNNs). Specifically, the authors argued that the receptive field (RF) of each convolutional filter should be not constrained in the small patch. Instead, it should also cover other locations beyond the local patch and also the single channel. Based on this motivation, the authors proposed a spatial shuffling layer which is aimed at shuffling the original feature responses. In the experimental results, the authors evaluated the proposed ss convolutional layer on CIFAR-10 and ImageNet-1k and compared with various baseline architectures. Besides, the authors further did some ablated analysis and visualizations for the proposed ss convolutional layer.

[Pros]

1. The authors proposed a new strategy for convolutional layers. The idea is borrowed from the biological domain, and then transformed to a spatial shuffling layer which can shuffle the feature response at each convolutional layer.

2. The authors performed experiments on both small-scale dataset (CIFAR-10) and large-scale data (CIFAR-100) for evaluations.

3. The authors further added some ablated analysis on the proposed model. Specifically, the authors visualize the receptive field which can be used in ss layer compared with the original convolutional layer, which indicates that ss layer can incorporate the global context at the very beginning.

[Cons]

1. The motivation behind the proposed ss layer is not explained very well. Though the authors mentioned that it is biologically inspired, I would not buy that since it is still a unclear phenomenon, and even it is true, using a randomized shuffling seems not align with the observations to some extent.

2. The paper is poorly written in general. The motivation behind the proposed method, and the presentation of method section part are cluttered much. In the model analysis in experiment section, the presentation and explanations are also vague and not clear to me.

3. The proposed model seems increase the baseline models' performance very marginally on all architectures. It is hard to say that it is because the shuffling layer enable the neurons to incorporate the global context information. Instead, it might just because the randomization which would increase the generalization ability of the trained model.

4. Finally, the comparison with previous models, such as SENet, ShuffleNet, etc are not systematically. I would like to see a more comprehensive summarization of the differences between the proposed ss layer and other architectures, because all of them are trying to incorporate more contextual information from other channels or locations.

[Summary]

Overall, I think the proposed ss layer is still a reasonable way to incorporate the contextual information in CNNs. However, the poor presentation and the weak experimental results and analysis make the paper overall a one under the bar of the venue. I would suggest the authors revise the paper with more well-motivated formula and more solid experiments and analysis in the next submission.

**Experience Assessment:**

I have published one or two papers in this area.

**Review Assessment: Checking Correctness Of Derivations And Theory:**

I carefully checked the derivations and theory.

**Review Assessment: Checking Correctness Of Experiments:**

I assessed the sensibility of the experiments.

**Review Assessment: Thoroughness In Paper Reading:**

I read the paper at least twice and used my best judgement in assessing the paper.

---

### Official Review · AnonReviewer2 · 2019-10-26
**Official Blind Review #2**

**Rating:** 3

**Review:**

This paper proposes a simple “spatial shuffling” operation for modifying CNNs
based on a permutation matrix created at initialization time.

The approach is motivate by a very high-level discussion of biological brains.
Improvements are claimed on Cifar10 but results are not near the state of the art.
The results do seem to improve incrementally over the previous vanilla results
on the particular architecures on which they are applied.

The authors dedicate some space to a qualitative analysis of why
the models improve although it is at best intuition-y.

In the end I’m left with inconclusive results, a weakly motivated story,
and a paper that despite exceeding the page limit by a page lacks
information density.


Minor:

Convolutional neural networks achieve … “elegant performance in computer vision tasks”
>>> 	wrong adjective.



**Experience Assessment:**

I have published one or two papers in this area.

**Review Assessment: Checking Correctness Of Derivations And Theory:**

I assessed the sensibility of the derivations and theory.

**Review Assessment: Checking Correctness Of Experiments:**

I assessed the sensibility of the experiments.

**Review Assessment: Thoroughness In Paper Reading:**

I read the paper at least twice and used my best judgement in assessing the paper.

---

### Decision · Program_Chairs · 2019-12-19

**Decision:**

Reject

**Comment:**

The paper is well-motivated by neuroscience that our brains use information from outside the receptive field of convolutive processes through top-down mechanisms. However, reviewers feel that the results are not near the state of the art and the paper needs further experiments and need to scale to larger datasets.